# Change in the Strategy of Embryo Selection with Time-Lapse System Implementation—Impact on Clinical Pregnancy Rates

**DOI:** 10.3390/jcm10184111

**Published:** 2021-09-12

**Authors:** Lisa Boucret, Léa Tramon, Patrick Saulnier, Véronique Ferré-L’Hôtellier, Pierre-Emmanuel Bouet, Pascale May-Panloup

**Affiliations:** 1Reproductive Biology Unit, Angers University Hospital, 49000 Angers, France; Lea.Tramon@chu-angers.fr (L.T.); VeFerre@chu-angers.fr (V.F.-L.); pamaypanloup@chu-angers.fr (P.M.-P.); 2Department of Biostatistics and Methodology, Angers University Hospital, 49000 Angers, France; Patrick.Saulnier@chu-angers.fr; 3Department of Reproductive Medicine, Angers University Hospital, 49000 Angers, France; PierreEmmanuel.Bouet@chu-angers.fr; 4MITOVASC, SFR ICAT, Inserm, CNRS, Angers University [Angers University Hospital], 49000 Angers, France

**Keywords:** embryo selection, time-lapse imaging, morphokinetic assessment, implantation, clinical pregnancy rate

## Abstract

Time-lapse systems (TLS) and associated algorithms are interesting tools to improve embryo selection. This study aimed to evaluate how TLS and KIDScore™ algorithm changed our practices of embryo selection, as compared to a conventional morphological evaluation, and improved clinical pregnancy rates (CPR). In the study group (year 2020, *n* = 303 transfers), embryos were cultured in an EmbryoScope+ time-lapse incubator. A first team observed embryos conventionally once a day, while a second team selected the embryos for transfer based on time-lapse recordings. In the control group (year 2019, *n* = 279 transfers), embryos were selected using the conventional method, and CPR were recorded. In 2020, disagreement between TLS and the conventional method occurred in 32.1% of transfers, more often for early embryos (34.7%) than for blastocysts (20.5%). Irregular morphokinetic events (direct or reverse cleavage, multinucleation, abnormal pronuclei) were detected in 54.9% of the discordant embryos. When it was available, KIDScore™ was decreased for 73.2% of the deselected embryos. Discordant blastocysts mainly corresponded with a decrease in KIDScore™ (90.9%), whereas discordant Day 3 embryos resulted from a decreased KIDScore™ and/or an irregular morphokinetic event. CPR was significantly improved in the TLS group (2020), as compared to the conventional group (2019) (32.3% vs. 21.9%, *p* = 0.005), even after multivariate analysis. In conclusion, TLS is useful to highlight some embryo development abnormalities and identify embryos with the highest potential for pregnancy.

## 1. Introduction

Embryo selection for transfer is a crucial step in the Assisted Reproductive Technology (ART) process and remains dependent on the morphological assessment of embryos. In the last few years, the time-lapse system (TLS) has become a promising tool to improve embryo selection and promote elective single embryo transfer (eSET) widely used to reduce multiple pregnancies after in vitro fertilization (IVF). The introduction of time-lapse incubators in IVF laboratories allows for continuous observation of embryo development without disruption of the controlled and stable culture conditions. Additional non-invasive criteria regarding embryo growth and development can be screened with TLS that snapshot morphological assessments could miss, such as altered morphokinetics, abnormal embryo cleavage (i.e., direct [1,2,3,4,5] or reverse cleavage [3,6,7]), and multinucleation [8]. Key timing and duration events during early embryo development are shown to be related to blastocyst formation [9,10,11,12,13], implantation [8,14,15], and live births [15,16], and several studies have built algorithms based on these parameters to select embryos with the highest implantation potential [13,14,17,18,19,20,21,22]. However, the generalizability of algorithms still remains questionable due to the influence of confounding factors on morphokinetic variables, such as patients’ characteristics (age, ovarian reserve, female body mass index, smoking habits, infertility indication, etc.), stimulation protocol, embryo culture techniques (fertilization method, culture media, gas composition), and embryo status (embryo quality and ploidy) [23]. Various authors report a lack of reproducibility between algorithms [24] or centers [25] and emphasise the need for either external validation of the models [26] or development of in-house algorithms based on the laboratory’s own data [27]. Nonetheless, the KIDScore™ (Vitrolife, Västra Frölunda, Sweden), a predictive model designed for EmbryoScope based on a large set of Known Implantation Data (KID) originating from a wide range of IVF clinics, would be universal and independent of culture conditions and fertilization methods [17]. Considering IVF outcomes, it remains controversial whether TLS can help improve pregnancy and live birth rates. Some studies observed significantly higher clinical pregnancy [18,28,29,30] and live birth rates [30,31,32,33,34] with TLS in comparison with conventional incubators and standard morphological evaluation, while other investigators found no significant differences between the two systems [8,35,36,37,38]. In addition, changes in the strategy employed for ranking embryos when using TLS have been minimally explored [39]. In this context, the present study aims to evaluate how time-lapse technology and KIDScore™ use have changed our current practices of embryo selection for transfer, in comparison with the conventional morphological evaluation, and contributed to improving pregnancy outcomes in IVF.

## 2. Materials and Methods

### 2.1. Study Design and Population

This was a monocentric ambispective study (prospective and retrospective) conducted at the reproductive medicine center of Angers hospital (France). The time-lapse incubator (EmbryoScope+, Vitrolife, Västra Frölunda, Sweden) has been available in routine procedures in the laboratory since January 2020 and the team had already been trained in its use. Since then, embryos have been selected for transfer using time-lapse technology. The main objective of this study was to evaluate the effect of the imaging system (time-lapse imaging versus conventional imaging) on the choice of embryos for uterine transfer. The secondary outcome was to compare the clinical pregnancy rates between the year 2020 (time-lapse imaging) and the year 2019 (conventional incubation). IVF cycles were suspended between 19 March and 16 June due to the Covid-19 pandemic. In order to standardize the 2 periods, we excluded cycles that occurred between 11 March and 17 June 2019.

We included all patients on the condition that they were between 18 and 43 years old (French standard age requirements for IVF), and they were not opposed to the processing of their data for research. Patients who had no fresh transfer due to the absence of embryo at cleavage or blastocyst stage or due to the total freezing of the cohort were excluded.

### 2.2. Controlled Ovarian Stimulation

Patients were treated by controlled ovarian hyperstimulation according to standardized clinical protocols using the gonadotropin-releasing hormone (GnRH) antagonist protocol (90% of cases) or the long GnRH agonist protocol (10% of cases). Briefly, the GnRH antagonist protocol consisted of a daily administration of recombinant or urinary gonadotropins, and premature ovulation was prevented with GnRH antagonist. The long GnRH agonist protocol consisted of mid-luteal pituitary down-regulation, using a daily dose of GnRH agonist (triptorelin 0.1 mg/day) associated with a recombinant or urinary gonadotropin administration. The regimen used and the dose of gonadotropins was adjusted according to age, body mass index (BMI), and ovarian reserve check-up. Ovarian response was monitored using ultrasonographical and blood measurements. Oocytes were retrieved transvaginally 36 h after administration of 5000 IU of recombinant human chorionic gonadotropin (rhCG), which was administered when at least 4 leading follicles were of ≥17 mm diameter.

### 2.3. Fertilization and Embryo Culture

Semen was collected and analyzed following the World Health Organization (WHO) guidelines [40]. Semen parameters were classified as abnormal if the prewash total motile sperm count (TMSC) was <5 million. Women underwent IVF with or without intracytoplasmic sperm injection (ICSI) according to the center’s guidelines. We performed ICSI at 400 magnification using an Integra Ti™ Micromanipulator (Cooper Surgical Company, Trumbull, CT, USA). In the conventional IVF cases, the oocytes were inseminated with 100,000 motile spermatozoa in 1 mL of FertiCult^®^ IVF medium (FertiPro, Beernem, Belgium). During the year 2019, oocytes after fertilization were cultured in Global medium^®^ (LifeGlobal, Målov, Denmark) with 10% of Human Serum Albumin (HSA) (Cooper-Surgical, Måløv, Denmark) at 37 °C in a humidified atmosphere containing 6% CO_2_ and 7% O_2_. Embryo morphology was conventionally assessed once a day until transfer. During 2020, embryos were cultured in an EmbryoScope+ time-lapse incubator (Vitrolife, Västra Frölunda, Sweden) in HSA-supplemented Global medium^®^ at stable conditions of 5% O_2_ concentration, 6% CO_2,_ and 37 °C. The time-lapse acquisition was set at 10-min intervals in 11 focal planes. Analysis was performed using a software developed for Embryoscope (EmbryoViewer software; Vitrolife, Västra Frölunda, Sweden). A first team, including at least one senior, ranked the embryos according to the conventional morphology assessment, whereas a second team (with at least one senior) selected embryos independently, using morphokinetic data of the EmbryoScope and the KIDScore results. The first team looked at time-lapse images at specific key time points without removing slides from TLS. The following information was collected: presence or absence of 2 pronuclei (at 17 +/− 1 h post-fertilization), early cleavage (at 26 +/− 1 h post-ICSI, 28 +/− 1 h post-IVF), multinucleation, number and symmetry of blastomeres, percentage of fragmentation (at 44 +/− 1 h and 68 +/− 1 h post-insemination), and blastocyst formation (at 116 +/− 2 h) [41]. Morphology was assessed with ESHRE classification for the cleavage stage embryos and with Gardner’s classification for blastocysts [41]. The second team analyzed the time-lapse recordings of embryo development. Data concerning morphokinetic parameters required for the application of the KIDScore™ (number of pronuclei, tPNf, t2, t3, t4, t5, t8, tB, ICM, and TE) and morphological parameters (abnormal pronuclei, multinucleation at the 2-cell stage, fragmentation, blastocyst collapse) were collected. Irregular events in embryo development such as direct cleavage (a single blastomere divides directly from 1 to 3 cells in less than 5 h) and reverse cleavage (a blastomere is re-absorbed after cleavage) were monitored according to the description by Rubio et al. [1] and Liu et al. [6]. All embryos were annotated according to the current guidelines [23,42] and scored using D3 or D5 KIDScore™. During this period, the selection of embryos for transfer was based on the data of the second team (time-lapse imaging).

### 2.4. Embryo Transfer and Freezing

Luteal phase support was initiated after retrieval with oral dydrogesterone (30 mg/day), and embryos were transferred into the uterus 2, 3, or 5 days after oocyte retrieval. The number of embryos selected for transfer and the day of transfer was chosen according to the patient’s age, parity, medical conditions, cycle rank, and embryo quality. All remaining good-quality embryos were cryopreserved by vitrification according to the practice protocols established in the center. Biological pregnancy was diagnosed by a serum bhCG level above 100 IU/L, which was tested 14 days after oocyte retrieval. A vaginal ultrasound was performed at 7 weeks gestational age (GA) to confirm a clinical pregnancy.

### 2.5. Data Collected

The demographic and baseline clinical information collected included the following data: age, BMI, tobacco use, stimulation treatment (protocol, type of gonadotropins and total dose of follicle-stimulating hormone (FSH) administered), causes of infertility (Diminished Ovarian Reserve (DOR), Polycystic Ovary Syndrome (PCOS), endometriosis, tubal factor) as well as the results of the ovarian reserve assessment (basal FSH, LH, and Estradiol, Antral Follicle Count (AFC) and anti-Müllerian hormone (AMH)). The following data about fertilization and embryo culture were also collected: sperm quality of the partner, insemination method (IVF or ICSI), number of oocytes and embryos obtained, number of embryos transferred, and day of transfer.

### 2.6. Outcome Measures

In a first part, we investigated the contribution of continuous monitoring provided by TLS and KIDScore™ result on the embryo selection process (2020). We studied the proportion of transfers for which we observed disagreement between the two selection methods. Disagreement was defined as complete when 1 embryo in case of SET or 2 embryos in case of DET were different between the 2 embryo selection methods. Disagreement was defined as partial in the other case (only 1 embryo different in the case of DET). We analyzed the reasons that led to a potential difference in the choice of embryos, such as KIDScore^TM^ result, and/or atypical cleavage features detected with time-lapse imaging. In the second part, we compared the clinical pregnancy rates between transfers that occurred in 2019 (conventional observation) and 2020 (time-lapse observation), defined as the presence of an intrauterine sac with a heartbeat visible on ultrasound at 7 weeks GA.

### 2.7. Statistical Analysis

Continuous variables were described with mean values and standard deviations and were compared using Student’s *t*-test. Categorical variables were summarized with their counts and percentages and were compared using Chi-square test. Two-tailed *p*-values < 0.05 were considered significant. Variables with a statistically significant association in univariate analysis were included in a multivariable logistic regression model. The odds ratio (OR) of variables that were associated with clinical pregnancies were provided with 95% confidence interval (95% CI) and statistical significance. All statistical analyses were performed using SPSS v15.0 (SPSS, Inc, Chicago, IL, USA) or R software [43].

### 2.8. Ethical Approval

Patients received an information letter describing the purpose of the study, the right to object to the processing of their data or to withdraw their consent at any time without any explanation or consequence. The study was conducted in accordance with the Declaration of Helsinki, and the protocol was approved by the Ethics Committee of the Angers University Hospital, France (Number 2020/72). All data were anonymously collected from the local database, in accordance with the French National Commission for Information and Liberties (ar20-0072v0).

## 3. Results

Among the 362 oocyte retrievals performed in 2019, 83 cycles were excluded (21 fertilization failures, 28 embryo culture failures, and 34 freeze-all strategies). As a result, 279 fresh embryo transfers were carried out using daily conventional embryo observation. During 2020, 401 egg retrievals were performed, including 32 fertilization failures, 31 embryo culture failures, and 35 freeze-all strategies. Accordingly, in 303 cycles, embryos were graded both by conventional observation and TLS. Then, 303 transfers were carried out based on TLS selection alone (Figure 1).

### 3.1. Daily versus Time-Lapse Observations

#### 3.1.1. Concordance between the Selection Methods

Considering the 303 transfers performed in 2020, embryo selection performed with TLS on the one hand and a conventional morphological grading system, on the other hand, was different for 77 transfers. It means that for 77 transfers, at least one embryo considered as the best quality for transfer and actually selected would not have been chosen if we had used conventional morphological observation instead of morphokinetic assessment and KIDScore™ result. In fact, there was no real choice to be made for 63 transfers (20.8%), since only one (in case of SET) or two embryos (in case of Double Embryo Transfer (DET)) were obtained in these cases. Consequently, there was a 32.1% disagreement (77/240) between TLS and the conventional morphology grading system. In detail, 17.1% partial disagreement (41/240) and 15% complete disagreement (36/240) were found between the two embryo selection methods.

#### 3.1.2. Concordance According to the Day of Transfer

We performed a subgroup analysis, taking into account the day of embryo transfer. Disagreement was most found for early embryo transfers (34.7%) compared to blastocyst transfers (20.5%), but this difference did not reach statistical significance (*p* = 0.07).

#### 3.1.3. Categories of Morphokinetic Anomalies

Among the 77 transfers corresponding with a change of embryo selection based on at least one embryo discordance, there were 5 DET with complete disagreement. As a result, 82 embryos were involved in a change of selection strategy. Irregular morphokinetic events (direct or reverse cleavage, multinucleation, abnormal pronuclei) were detected in 54.9% of these (Table 1).

As a KIDScore™ is available only for transfers performed on Day 3 or 5, 53 transfers (56 embryos) were included in the following analysis. The reasons that led to discrepancies between the two embryo selection methods are described in Table 2. A lower KIDScore™ was observed for 73.2% (41/56) of the deselected embryos.

A large majority of discordant blastocysts were connected with a decrease in KIDScore™ (90.9%), whereas discordant embryos on Day 3 were distributed between a decrease in KIDScore™ with or without abnormal cleavage (51.1% and 17.8% respectively), multinucleation (8.9%) and subjective judgment (22.2%).

### 3.2. Comparison between TLS and Conventional Incubation and Selection on IVF Outcomes

The second part of this study aimed to evaluate whether the use of TLS for embryo incubation and selection could improve pregnancy rates. For this purpose, we compared reproductive outcomes obtained in 2020 (incubation and selection with TLS) with those obtained in 2019 (conventional incubation and selection).

#### 3.2.1. Patient Characteristics

Baseline clinical characteristics and stimulation parameters are summarized in Table 3. Baseline patients’ characteristics were comparable between the two groups regarding age, tobacco use, and cause of infertility, except for BMI, which was significantly higher in 2019 (24.7 ± 4.9 vs. 23.8 ± 4.9, *p* = 0.03). Both groups showed no difference in the ovarian reserve markers (AMH, basal FSH and AFC). Abnormal semen parameters and insemination method did not differ between the two groups (*p* = 0.4 and *p* = 0.1, respectively). The total dose of FSH used for ovarian stimulation was significantly higher in 2019 (2 509 ± 1 102 IU) compared with 2020 (2 215 ± 870 IU) (*p* < 0.001). In 2020, patients were more likely to have undergone an agonist protocol for ovarian stimulation (15.5% vs. 9.3%, *p* = 0.02). There was no significant difference in the type of gonadotropin used (*p* = 0.1).

#### 3.2.2. IVF Parameters and Reproductive Outcomes

The IVF parameters and reproductive outcomes are summarized in Table 4. The mean number of oocytes per patient did not differ between the two groups (*p* = 0.3). Similarly, we did not document any significant difference when considering the mean number of embryos obtained (*p* = 0.4). The proportion of single embryo transfer was comparable between the two groups (*p* = 0.2). Conversely, more embryo transfers were performed on Day 2 in 2019 (60.9%), when compared to 2020 (33.3%) (*p* < 0.001). Biological pregnancy rates (34.3% vs. 24.0%, *p* = 0.006), implantation rates (23.1% vs. 16.4%, *p* = 0.01), and clinical pregnancy rates (32.3% vs. 21.9%, *p* = 0.005) were significantly improved in 2020 (time-lapse observation) compared to 2019 (conventional observation).

#### 3.2.3. Multivariate Analysis

A logistic regression was performed to adjust for baseline differences in BMI, total FSH dose administered, stimulation protocol, day of transfer, and TLS use. Results are presented in Table 5. When controlling for confounding factors, time-lapse technology remained associated with higher clinical pregnancy rates (OR = 1.57; 95% CI (1.05–2.36); *p* = 0.029). The total dose of FSH administered was also significantly different between the two groups in the multivariate analysis, but with a marginal effect (OR = 0.99; 95% CI (0.99–1.00); *p* = 0.016). We undertook a backward elimination procedure and dropped TLS from our logistic regression model. The best-fit model according to the Akaike Information Criterion (AIC) was the model with TLS: AIC = 651 in the model with TLS, and AIC = 654 in the model without TLS (Likelihood Ratio Test *p*-value = 0.03).

## 4. Discussion

Embryo selection is one of the most critical tasks in IVF laboratories and requires focusing particular attention to optimize the chances of implantation, encourage the practice of SET, and shorten the time to live birth. Over the last few years, time-lapse technology has provided a large additional amount of morphokinetic data in order to improve embryo selection without disrupting culture conditions through repeated removals from the incubator. It has contributed to identifying some morphokinetic parameters related to embryo viability [9,10,11,12,13,44] and implantation [8,14,15], in particular through the development of algorithms. The main objective of this study was to evaluate the impact of time-lapse assessment and KIDScore™ use on the embryo selection strategy compared with a standard morphological evaluation. We observed a significant change in the selection of embryos for transfer. For 34.7% of transfers, early-stage embryos were deselected and replaced by embryos with a better KIDScore™ or devoid of abnormal cleavage features. This change occurred more rarely for blastocysts, concerning 20.5% of transfers. These results are consistent with a previous publication that showed an agreement between the embryologist’s choice and KIDScore™ at Day 5 of 78% or 61%, depending on the version of KIDScore™ used [39]. In contrast, a more frequent disagreement was reported in a study conducted by Kovacs et al. [36], as a different embryo was selected for transfer in 50% of cycles when the time-lapse evaluation was applied instead of conventional morphology alone. In contrast with our population, the two aforementioned studies included only single blastocyst transfers and only good prognosis patients in the Kovacs’ study.

Discordant embryos on Day 3 were mainly distributed between abnormal cleavage and lower KIDScore™ result, whereas divergent embryos on Day 5 were overrepresented by a decrease in KIDScore™. This difference is consistent with the hypothesis that atypical cleavage features could have severe adverse effects on embryo development, with a minority of these embryos developing until the blastocyst stage. Time-lapse technology enabled us to highlight the presence of irregularities such as multi-nucleation and abnormal embryo cleavage (direct or reverse cleavage), which were not always detectable with conventional daily observations. These morphokinetic defects are underestimated with the standard evaluation, as these are detected only if they match the time of observation. These data are, however, still crucial for embryo selection, because they are believed to be linked with the embryo fate. The KIDScore™ enabled to detect the majority of embryo development abnormalities, although there were some exceptions, such as multinucleation, which had no effect on morphokinetics.

The prevalence of such morphokinetic abnormalities has already been described in the literature, as well as their impact on blastocyst development and implantation. Nonetheless, the impact of cleavage abnormalities and/or KIDScore™ result on the embryo deselection process has not been greatly considered before. Direct cleavage was found to be negatively predictive of developmental capacity to blastocyst [3,4,5] and implantation potential [1,2,5]. Regarding reverse cleavage, Liu et al. reported that the proportion of good quality on Day 3, as well as implantation rates, were statistically lower in embryos with this feature [6]. Description of multinucleation by time-lapse monitoring was also associated with decreased rates of implantation [8,45,46], clinical pregnancy [45], and live birth [46,47]. Finally, blastocyst collapse, which was found in almost 20% of embryos, would be negatively linked to the implantation process [48].

The secondary outcome was to evaluate the potential impact of the embryo selection strategy on the IVF outcomes. We observed a significant increase in clinical pregnancy rates between 2019 (conventional observation) and 2020 (time-lapse imaging). In the multivariate analysis, we showed that TLS was the only factor positively associated with clinical pregnancy rates. Our data chimed with other studies that found a significant positive correlation between KIDScore™ and implantation [39] or live birth [16]. Nonetheless, culture conditions were also improved with the time-lapse system, and this study was not able to assess the contribution of time-lapse parameters independently from the incubator part. However, some previous studies investigating the impact of the culture system alone found no superiority of the EmbryoScope [35,49,50], but without clear consensus [31]. Conversely, studies that assess the influence of the embryo-scoring algorithm when all embryos are cultured in the same incubator also show inconsistent results [8,29,36,51,52].

Most algorithms used to date are based on single-center embryo cohorts. Zaninovic et al. investigated whether standard morphokinetic parameters were equally distributed between two different clinics [53]. Unfortunately, the predictive value of these variables on blastocyst formation and implantation potential was different between the two centers. Indeed, morphokinetic variables appear to be strongly influenced by different center-specific parameters, such as culture conditions, annotation method, patients’ characteristics, and stimulation protocols. Any predictive parameter or algorithm could lose its value if externally applied and should be subject to a preclinical validation based on the specific intrinsic conditions of each laboratory before routine application. This dilemma was also underlined by other authors [24,26,27,54], limiting the development of universal predictive models. However, the KIDScore™ algorithm could be considered as an exception, as it was developed on a known implantation database of 3275 embryos originating from 24 clinics [17]. The KIDScore™ algorithm presented a comparable Area Under the Curve (AUC) when tested for different gas composition or fertilization methods. Moreover, the KIDScore™ algorithm showed the highest agreement with the majority of 10 experienced embryologists when compared with 6 other tested algorithms [24]. However, in turn, as the KIDScore™ was designed as a deselection algorithm, it was also the algorithm where the embryologist most often had to make the final decision, when more than one embryo was deemed to be the best for transfer.

This lack of reproducibility could explain why there are still conflicting data regarding the clinical outcomes with time-lapse systems. Indeed, some authors demonstrate that time-lapse technology and morphokinetic scores could improve implantation [18,29] and clinical pregnancy rates [28,29,32,55] compared with conventional methods, while others found no significant difference [8,36,37]. Differing results persist in meta-analyses: Pribenzky’s study [33], based on 1637 randomized patients, showed reduced early pregnancy loss, higher ongoing pregnancy and live birth rates with time-lapse culture and morphokinetic embryo selection. Magdi et al. emphasized a 9% improvement in a live birth with TLS [34]. In contrast, the updated Cochrane review [56], including 2955 patients, as well as two other meta-analyses [38,57], reported insufficient evidence to support that TLS is superior to conventional methods for human embryo incubation and selection.

In this quest for reliability and reproducibility, recent studies explored novel independent morphokinetic parameters developed as biomarkers of embryo quality and implantation. For instance, Coticcio et al. listed several time intervals during the fertilization process that were strongly associated with embryo quality on Day 3, revealing the importance of neglected phenomena such as the fertilization cone, cytoplasmic wave, and cytoplasmic halo [58]. Barberet et al. showed that central pronuclei position at the time of juxtaposition was associated with increased live birth rates [47]. This qualitative parameter could be relatively less sensitive to inter-laboratory variations and easier to reproduce than quantitative parameters. At the morula stage, partial compaction with excluded or extruded cells seems to have an adverse effect on blastulation, blastocyst morphology, and live births [59]. At the blastocyst stage, trophectoderm cell cycle length and blastocyst expanded diameter were described as independent variables that could discriminate implanted from non-implanted embryos [60].

The introduction of innovative technologies in embryology laboratories gives great attention to improving embryo selection and IVF outcomes. Deep learning could represent a promising tool integrating all data obtained from time-lapse technology. Machine learning could potentially consider the subtle effects of confounders and the complex relationships that exist between various morphokinetic parameters and specific patient and treatment factors that the human eye and the currently available software are not able to detect [61]. In a recent study, Khosravi et al. used convolutional neural networks to predict blastocyst quality, with an AUC of >0.98 and a robust performance for inter-center application [62]. Tran et al. developed a deep learning model able to predict clinical pregnancies, with an AUC of 0.93 (95% CI 0.92–0.94) [63]. With regard to preimplantation genetic screening (PGS), Rocafort et al. combined next-generation sequencing (NGS) with automated TLS for embryo selection [55]. They concluded, similar to others [64,65,66,67], that TLS could not substitute PGS to predict blastocyst ploidy, but they also found that implantation and pregnancy rates were significantly improved when combining PGS with TLS, in comparison with PGS alone. A recent study confirmed the clinical relevance of combining the two aforementioned techniques [68]. In another field, Alegre et al. proposed an algorithm combining morphokinetic parameters with the oxidative status of the spent embryo culture medium to improve embryo selection and IVF outcomes [69]. However, non-invasive techniques based on metabolomic or proteomic approaches appear to be hardly applicable in current practice.

This study should be considered in light of its strengths and limitations. First, this was a single-center, exhaustive study without selection bias. We included patients with various prognoses, heterogeneous regarding baseline characteristics, stimulation protocol, oocyte yield, insemination method, and embryo transfer stage in order to validate embryo selection criteria that could be further applied to all patients of our center. Secondly, the comparison between TLS and conventional morphology assessment was prospective, with real-time embryo monitoring. All embryos were assessed by two embryologists, and we used a semi-automated annotation software for the time-lapse system in order to reduce inter-operator variability [9]. However, some limitations should be pointed out. Because of the Covid-19 pandemic, we were constrained to suspend transfers between March and June 2020. We compared two similar periods between the two years to avoid a potential seasonality bias, even though no significant difference of seasonal variations in the outcomes of IVF was observed in the literature [70,71]. A pending question concerned pregnancies after double embryo transfers, where a 50% agreement was found. It was not possible to discern whether implantation was related to the concordant or discordant embryo. There was also no way to know the potential outcome if the embryos that we deselected with time-lapse technology had been transferred. However, such information was not useful to evaluate the global effect of the change in embryo selection method between the two different periods. These are questions that would be interesting to explore in further studies but that the study design did not intend to answer. A limitation to the second part of this study was that, due to differences in culture conditions, the exact role of morphokinetic selection in improving outcomes could not be determined. We also acknowledge that embryo quality is not the only determinant related to IVF success and that endometrial receptivity plays a crucial role in successful implantation and progression to live birth. Finally, although live birth is considered as the best meaningful endpoint, clinical pregnancies were recorded because deliveries that occurred during the second part of this study were not necessarily known.

In conclusion, this study supports the growing evidence for the clinical benefit of culture and examination with TLS. The introduction of TLS in current practices changed our embryo selection strategy in about one-third of cases through continuous monitoring of embryos and the use of the KIDScore™. This change of practices contributed to a significant increase in clinical pregnancy rates. Unfortunately, this study was not designed to determine whether the difference observed in IVF results was related only to the selection method or also to the culture conditions. Further large-scale multicenter prospective studies, based exclusively on single embryo transfers, are still needed to draw firm conclusions about the clinical value of TLS.

## Figures and Tables

**Figure 1 jcm-10-04111-f001:**
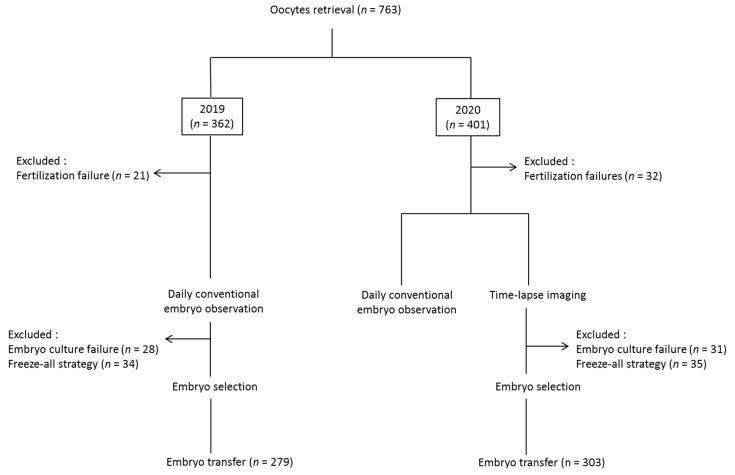
Flow chart of this study.

**Table 1 jcm-10-04111-t001:** Types of anomalies detected with TLS for deselected embryos. Values are shown as *n* (%).

	*n* (%)
Direct cleavage	23 (28%)
Reverse cleavage	4 (4.9%)
Multinucleation	17 (20.7%)
Abnormal pronuclei	1 (1.2%)
None	37 (45.1%)

**Table 2 jcm-10-04111-t002:** Reasons for a change in the embryo selection process for transfers on day 3 or 5 (KIDScore™ available). Values are shown as *n* (%).

	*n* (%)
Detected abnormal cleavage associated with a decrease in KIDScore™	26 (46.4%)
Lower KIDScore™ only	15 (26.8%)
Multinucleation	5 (8.9%)
Subjective judgment	10 (17.9%)

**Table 3 jcm-10-04111-t003:** Baseline characteristics and stimulation parameters in the conventional observation group (2019), and the time-lapse observation group (2020). Values are shown as mean ± SD or *n* (%).

	Conventional Observation Year 2019 (*n* = 279)	Time-Lapse Observation Year 2020 (*n* = 303)	*p*-Value
Female age (year)	33.1 ± 4.6	33.7 ± 4.7	0.1
BMI (kg/m^2^)	24.7 ± 4.9	23.8 ± 4.9	0.03
Tobacco use			
- Never	207 (74.2%)	227 (75.2%)	0.1
- Current	48 (17.2%)	37 (12.3%)	
- Former	24 (8.6%)	38 (12.6%)	
Cause of female infertility			0.2
- None	128 (45.9%)	125 (41.3%)	
- DOR	53 (19.0%)	53 (17.5%)	
- PCOS	34 (12.2%)	47 (15.5%)	
- Endometriosis	22 (7.9%)	23 (7.6%)	
- Tubal	18 (6.5%)	13 (4.3%)	
- Mixed	24 (8.6%)	42 (13.9%)	
AMH (ng/mL)	3.0 ± 2.8	3.1 ± 2.5	0.7
Basal FSH (IU/L)	7.6 ± 2.9	7.7 ± 3.1	0.9
Basal LH (IU/L)	5.6 ± 2.6	5.8 ± 2.8	0.3
Basal E2 (pg/mL)	43.2 ± 27.5	41.7 ± 21.7	0.5
AFC	19.4 ± 12.7	20.6 ± 12.1	0.2
Semen parameters			0.4
- Normal	183 (65.6%)	208 (68.6%)	
- TMSC < 5 millions	96 (34.4%)	95 (31.4%)	
Stimulation Protocol			0.02
- GnRH Antagonist (%)	253 (90.7%)	256 (84.5%)	
- GnRH Agonist (%)	26 (9.3%)	47 (15.5%)	
Gonadotropin type			0.1
- FSH	175 (62.7%)	208 (68.6%)	
- FSH + LH	104 (37.3%)	95 (31.4%)	
Total dose of FSH (IU)	2 509 ± 1 102	2 215 ± 870	<0.001

**Table 4 jcm-10-04111-t004:** IVF cycle characteristics and reproductive outcomes according to the embryo selection method. Values are shown as mean ± SD or *n* (%).

	Conventional Observation Year 2019 (*n* = 279)	Time-Lapse Observation Year 2020 (*n* = 303)	*p*-Value
Insemination method			0.1
- IVF	122 (43.7%)	146 (48.2%)	
- ICSI	153 (54.8%)	147 (48.5%)	
- IVF + ICSI	4 (1.4%)	10 (3.3%)	
Oocytes retrieved	10.7 ± 5.6	11.2 ± 6.3	0.3
Embryos obtained	5.5 ± 3.5	5.7 ± 3.7	0.4
Embryo transfer			
- Single embryo transfer	106 (38.0%)	130 (42.9%)	0.2
- Double embryo transfer	173 (62.0%)	173 (57.1%)	
Stage at embryo transfer			
- Day 2	170 (60.9%)	101 (33.3%)	<0.001
- Day 3	92 (33.0%)	157 (51.8%)	
- Blastocyst	17 (6.1%)	45 (14.9%)	
Biological pregnancy	67 (24.0%)	104 (34.3%)	0.006
Implantation	74 (16.4%)	110 (23.1%)	0.01
Clinical Pregnancy	61 (21.9%)	98 (32.3%)	0.005
Single/Twin Pregnancy	48/13	86/12	

**Table 5 jcm-10-04111-t005:** Likelihood of clinical pregnancy presented as odds ratios (95% confidence intervals).

Characteristic	OR	95% CI	*p*-Value
BMI	0.99	0.95, 1.03	0.6
Total dose of FSH	0.99	0.99, 1.00	0.016
Stimulation Protocol			
GnRH Agonist	1.0	—	
GnRH Antagonist	1.32	0.72, 2.52	0.4
Stage at embryo transfer			
Day 2	1.0	—	
Day 3	1.13	0.74, 1.72	0.6
Day 5	1.41	0.74, 2.62	0.3
Embryo selection method			
Conventional morphology (2019)	1.0	—	
Time-lapse observation (2020)	1.57	1.05, 2.36	0.029
OR = Odds Ratio, CI = Confidence Interval			

## Data Availability

The data presented in this study are available on request from the corresponding author.

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
