# Peer review of "Change in the Strategy of Embryo Selection with Time-Lapse System Implementation—Impact on Clinical Pregnancy Rates"

_jcm, 2021, doi:10.3390/jcm10184111_

Round 1
Reviewer 1 Report
As the authors themselves note several times, the main issue with the study is the different incubations conditions between 2019 and 2020. EmbryoScope incubator was used in 2020 with 5%Ox as compared to a different incubator with 7%Ox in 2019. Therefore, there is no way to know if the increase in clinical pregnancy rates in 2020 are due to the use of TLS and KIDScope or simply a difference in incubation. With this said, I think this is a well conducted study presented in a logical manner. There are some minor issues with referring to Tables (these appear as "error" on the version of the paper I reviewed). Focusing on the data regarding just 2020 with comparing embryo selection methods (TLS versus conventional morph grading), this is where I think some edits must be made. It is unclear what the authors really want the reader to know what was gained from that part of the study. It is known that TLS provides more info on the developing blastocyst, so the results provided are not really that novel.
Reviewer 2 Report
The paper presents a study comparing implantation rates for conventional incubation and morphological evaluation carried out in 2019 to time-lapse incubation and morphokinetic/morphological evaluation carried out in 2020.
The study group includes 303 transfers, and the control group includes 279 transfers. If no other factors than conventional incubation vs. time-lapse were changed between the two groups, this seems like a sufficient sample size for such a study. However, unfortunately other factors varied that could have influenced implantation rates just as much. Among these, there was a significant change in day of transfer, which at least partly can explain the difference in implantation rate. Also, the study group and control group were collected across different years, which could also influence the results.
In the multivariate analysis, a logistic regression model is applied to adjust for some of the confounding factors mentioned above. However, to isolate the effect of TLS, two separate models should be fitted - one including TLS and one without TLS. The two models could then be compared for significant difference.
Although the study question is relevant and of general interest, the limited sample size combined with many degrees of freedom (days of transfer D2/D3/D5, single/multi embryo transfer, incubation conditions, KIDScore versions D3/D5, ...) makes the conclusions questionable.
Below are some points that should be addressed in the paper, if revised:
- There are missing references throughout the manuscript ("Error! Reference source not found")
- L184: How are embryos incubated in TLS graded conventionally? Do embryologists look at time-lapse images at specific time points, or are the embryos removed from TLS to be inspected with traditional microscopes?
- L192: It is not clear what is meant by embryo selection being different between conventional/TLS grading. Does it mean that the ranking of embryos within a treatment was different? Please elaborate.
- L197: What is the difference between partial and complete disagreement?
- Table 4: Wrong percentages on "Implantation" row.
Round 2
Reviewer 2 Report
The authors have addressed all my questions and comments, and the revision has improved the paper considerably.
The addition of the backward elimination procedure comparing a logistic regression model with and without TLS, especially, is much appreciated. However, the analysis does not result in an evaluation for significant difference between the two models. The reader is left with their own judgement of whether AIC=651 is significantly better than AIC=654. To the best of my knowledge, it should be possible to estimate a P-value.
Regarding the limited sample size, the authors compare their sample size with studies included in the last Cochrane analysis. However, the Cochrane analysis only included RCTs. As such, these studies were not subject to selection bias (and potential historical bias between 2019 and 2020), and are thus not directly comparable.
Small corrections in revision:
- L207: "Its" should be "It"
- L281: "undetook" should be "undertook"
Author Response
We thank the reviewer for these relevant comments.
- As required, we compared the two models with a Likelihood Ratio Test (LRT).
P-value returned by LRT was 0.03. We modified the manuscript accordingly (Line 284).
- Small corrections in revision:
- L207: "Its" should be "It".
- L281: "undetook" should be "undertook".
We corrected the manuscript accordingly.